# Reactive Metamizole Metabolites Enhance the Toxicity of Hemin on the ATP Pool in HL60 Cells by Inhibition of Glycolysis

**DOI:** 10.3390/biomedicines8070212

**Published:** 2020-07-14

**Authors:** Deborah Rudin, Maurice Schmutz, Noëmi Johanna Roos, Jamal Bouitbir, Stephan Krähenbühl

**Affiliations:** 1Division of Clinical Pharmacology & Toxicology, University Hospital Basel, Spitalstrasse 21, 4031 Basel, Switzerland; deborah.rudin@unibas.ch (D.R.); maurice.schmutz@unibas.ch (M.S.); noemi.roos@unibas.ch (N.J.R.); jamal.bouitbir@unibas.ch (J.B.); 2Department of Biomedicine, University of Basel, Hebelstrasse 20, 4031 Basel, Switzerland; 3Swiss Centre for Applied Human Toxicology (SCAHT), University of Basel, Missionsstrasse 64, 4055 Basel, Switzerland

**Keywords:** metamizole, agranulocytosis, mechanism, glycolysis, mitochondria

## Abstract

Metamizole is an analgesic, whose pharmacological and toxicological properties are attributed to N-methyl-aminoantipyrine (MAA), its major metabolite. In the presence of heme iron, MAA forms reactive metabolites, which are toxic for granulocyte precursors. Since decreased cellular ATP is characteristic for MAA-associated toxicity, we studied the effect of MAA with and without hemin on energy metabolism of HL60 cells, a granulocyte precursor cell line. The combination MAA/hemin depleted the cellular ATP stronger than hemin alone, whereas MAA alone was not toxic. This decrease in cellular ATP was observed before plasma membrane integrity impairment. MAA/hemin and hemin did not affect the proton leak but increased the maximal oxygen consumption by HL60 cells. This effect was reversed by addition of the radical scavenger *N*-acetylcysteine. The mitochondrial copy number was not affected by MAA/hemin or hemin. Hemin increased mitochondrial superoxide generation, which was not accentuated by MAA. MAA decreased cellular ROS accumulation in the presence of hemin. In cells cultured in galactose (favoring mitochondrial ATP generation), MAA/hemin had less effect on the cellular ATP and plasma membrane integrity than in glucose. MAA/hemin impaired glycolysis more than hemin or MAA alone, and *N*-acetylcysteine blunted this effect of MAA/hemin. MAA/hemin decreased protein expression of pyruvate kinase more than hemin or MAA alone. In conclusion, cellular ATP depletion appears to be an important mechanism of MAA/hemin toxicity on HL60 cells. MAA itself is not toxic on HL60 cells up to 100 µM but boosts the inhibitory effect of hemin on glycolysis through the formation of reactive metabolites.

## 1. Introduction

Metamizole (dipyrone) is an analgesic and antipyretic drug that is used frequently in human and veterinary practice in some countries in Europe and South America but has been withdrawn from the market in other countries such as for instance Sweden, England, and the United States. In the postoperative situation, it is more effective than paracetamol and at least as effective as non-steroidal anti-inflammatory drugs [1,2], whereas data for the effectiveness in long-term treatments such as for osteoarthritis are lacking. Metamizole is a prodrug, which is transformed non-enzymatically to *N*-methyl-aminoantipyrine (MAA) already in the gastrointestinal tract [3]. MAA has a bioavailability close to 100% and is the main metabolite circulating in plasma [4,5].The analgesic activity of metamizole can be attributed mainly to the formation of MAA. The mode of action is not known with certainty but may be related to the inhibition of cyclooxygenase (COX) 2 [6].

Metamizole is generally well tolerated. It can cause anaphylaxis [7,8] and skin eruptions, which are considered to result from delayed allergic reactions [9,10]. The most severe adverse reaction is myelotoxicity, mostly in the form of neutropenia or agranulocytosis [11]. The frequency of this reaction is not exactly known and depends also on the method used for its determination; estimates range from 1:1500 of patients treated [12] to approximately 1 per million person days of use [13] and 0.5–9 per million inhabitants per year [14,15,16]. This adverse reaction may be caused by an allergic mechanism or by direct toxicity due to the formation of reactive metabolites. An allergic mechanism is supported by the fact that metamizole can cause skin eruptions [9,17] and liver injury [18,19] by a delayed type of drug allergy and by the finding that myelotoxicity may be related to a certain HLA constellation [20]. However, allergic features such as exanthema, eosinophilia, or lymphadenopathy are usually lacking in patients with metamizole-induced myelotoxicity [21,22], rendering an allergic mechanism less probable. MAA can react with the iron contained in heme and form reactive electrophilic metabolites [6,23], which suggests another mechanism for the observed myelotoxicity. Iron complexed in heme, the reduced form of hemin, is abundant in the bone marrow [24], enabling the formation of electrophilic MAA metabolites at the same location where neutrophil precursor cells are formed. We have shown recently that the toxicity of these MAA metabolites is restricted to neutrophil precursors in the promyelocyte and myelocyte stage of myelopoiesis, since promyelocytes and myelocytes have a low antioxidative capacity [25]. The observation that promyelocytes and myelocytes are the primary target of MAA-associated toxicity is in agreement with clinical findings, showing a stop in myelopoiesis at the promyelocyte and myelocyte stage in the bone marrow of affected patients [26].

The molecular mechanism of this potentially lethal adverse reaction is currently not known. Our previous studies showed that the combination of MAA with hemin is cytotoxic for promyelocytic HL60 cells [23,25]. These previous studies suggested that ATP depletion plays an important role, which may eventually lead to cell death. ATP can be generated by mitochondria and by glycolysis. Based on these considerations, we studied the effect of MAA in the absence and in the presence of hemin on the energy metabolism of HL60 cells. HL60 cells are a granulocyte precursor cell line in the promyelocyte maturation stage, which represents the maturation stage of neutrophil granulocytes that can be affected by metamizole.

## 2. Experimental Section

### 2.1. Chemicals and Cell Culture Reagents

*N*-methyl-4-aminoantipyrine, hemin, amiodarone, Triton X-100, dihydrorhodamine-123 (DHR-123), oligomycin, carbonyl cyanide 4-(trifluoromethoxy)phenylhydrazone (FCCP), rotenone, *N*-acetylcysteine (NAC), galactose, glucose and 2-deoxy glucose (2-DG) were purchased from Sigma–Aldrich (Buchs, Switzerland). Tetramethylrhodamine methyl ester (TMRM) was obtained from Invitrogen (Basel, Switzerland). RPMI-1640 medium, fetal bovine serum (FBS), and phosphate buffered saline (PBS) were purchased from GIBCO (Lucerne, Switzerland).

### 2.2. Time Course of Cellular ATP Depletion

To assess the time course of ATP decrease in HL60 cells treated with MAA/hemin, the cells were incubated with MAA and hemin, and cellular ATP was assessed at time points 0 h, 4 h, 8 h, 12 h, 16 h, 20 h, and 24 h. For the experiments, 2 × 10^5^ HL60 cells were seeded in 1 mL RPMI containing 10% FBS in a 24-well plate. The stock solution of MAA was prepared in DMSO and added to the cell solution at a final concentration of 100 µM. This concentration can be reached in serum of patients treated with metamizole [27,28]. Ferric iron (hemin) was added to the reaction solution at a final concentration of 12.5 µM. This concentration was based on our previous studies [23,25] and on the fact that free hemin concentrations erythrocytes can reach up to 20 µM [29]. Since hemin is not readily soluble in PBS, it was first dissolved in 10 mM NaOH as a 1 mM stock solution and then diluted to the final concentration. An equivalent concentration of NaOH was added to the vehicle control. The concentrations of MAA were chosen based on available pharmacokinetic data in humans and previously obtained results and lie within plasma concentration ranges reached after multiple dosages of metamizole [28]. Hemin concentrations were chosen based on previously obtained results and are around 1000 times lower than the physiological hemoglobin concentration, whereby one hemoglobin molecule contains four hemin molecules. We added 50 µL of each suspension-mixture in triplicate to a 96-well white clear bottom microplate (BD Biosciences, Franklin Lakes, NJ, USA) and incubated the plate for 0 h, 4 h, 8 h, 12 h, 16 h, 20 h, and 24 h at 37 °C, 5% CO_2_. The DMSO concentration did not exceed 0.1% in all incubations, including control incubations, as this DMSO concentration is not cytotoxic [30]. Triton X-100 was used at a final concentration of 0.1% as a positive control for plasma membrane toxicity. All experiments were performed in triplicates and repeated at least three times using different cell isolations.

To assess the cellular ATP content as a marker for cellular energy metabolism, the CellTiter-Glo^®^ luminescent assay (Promega Corporation, Madison, WI, USA) was used. For that, 50 µL assay buffer was added to 50 µL cell suspension prepared as described above and luminescence was measured with a Tecan Infinite pro 200 microplate reader (Tecan, Männedorf, Switzerland) after 10 min of incubation.

### 2.3. Time Course of Loss of Membrane Integrity

To assess the time course of the intactness of the plasma membrane of HL60 cells treated with MAA/hemin, 2 × 10^5^ cells were incubated with MAA and hemin and membrane toxicity was assessed at time points 0 h, 4 h, 8 h, 12 h, 16 h, 20 h, and 24 h. HL60 cells were treated with MAA and hemin as outlined above. To assess a potential loss of plasma membrane integrity, which is reflected by the release of adenylate kinase, the firefly luciferase system (ToxiLight^®^ BioAssay Kit, Lonza, Basel, Switzerland) was used. After incubation in the presence of the test compounds, 50 µL assay buffer were added to 50 µL cell suspension and luminescence was measured with a Tecan Infinite pro 200 microplate reader (Tecan, Männedorf, Switzerland) after 5 min of incubation.

### 2.4. Cellular Oxygen Consumption Rate (OCR)

HL60 cells were incubated with MAA and hemin in 75 cm^2^ tissue culture flasks (TPP Techno Plastic Products AG, Trasadingen, Switzerland) at a density of 2 × 10^5^ cells/mL. After incubation for 24 h in presence of test compounds, HL60 cells were washed with PBS and resuspended in sterile assay buffer (unbuffered RPMI-1640 containing glucose supplemented with 1 mM sodium pyruvate, pH 7.4) and plated (75,000 cells/well) on a Cell-Tak (Corning, Bedford, MA, USA) coated XF 96-well cell culture microplate (Seahorse Bioscience; Agilent Technologies, Santa Clara, CA, USA). The plate was incubated for 1 h in a CO_2_ free incubator at 37 °C before cellular bioenergetics of the cells was determined using the extracellular flux analyzer of the Seahorse XF 96 (Seahorse Bioscience). First, basal respiration was determined by measuring the OCR before addition of any inhibitors or activators. OCR linked to oxidative phosphorylation was determined by injection of oligomycin at a final concentration of 1 µM. Next, to determine the maximal respiration rate, the electron transport chain uncoupler FCCP was used at a final concentration of 2 µM. Finally, non-mitochondrial OCR was determined by injection of rotenone at a final concentration of 1 µM. Respiration rates were calculated after correction for non-mitochondrial OCR.

### 2.5. Mitochondrial Superoxide Generation

HL60 cells (2 × 10^5^ cells) were treated with MAA and hemin as outlined above. As a positive control, 50 µM amiodarone was used. After 24 h of treatment, the cells were washed with PBS and resuspended in 100 µL PBS containing 2.5 µM MitoSOX Red reagent (Invitrogen, Basel, Switzerland) in a clear v-bottom 96-well microplate (Corning, Bedford, MA, USA). MitoSOX Red selectively targets mitochondria and is oxidized by superoxide but not by other reactive oxygen species (ROS) to a fluorescent dye. It is therefore a marker for mitochondrial superoxide. After 10 min incubation in the dark at room temperature, fluorescence was measured by flow cytometry with a CytoFLEX flow cytometer (Beckmann Coulter, Indianapolis, IN, USA) and data were assessed using FlowJo software 10.08 (Tree Star, Ashland, OR, USA).

### 2.6. Total Cellular ROS Production

To assess the total cellular ROS content, 2 × 10^5^ HL60 cells were treated with MAA and hemin as mentioned above. As positive control, 50 µM amiodarone was used. After 24 h incubation, the cells were washed with PBS and resuspended in 200 µL X-Vivo serum-free medium (Lonza, Verviers, Belgium) containing 1 µM dihydrorhodamine 123 (DHR-123) in a black flat bottom 96-well microplate (Corning, Bedford, MA, USA). DHR-123 is an uncharged, non-fluorescent molecule which diffuses passively across membranes. After oxidation by ROS to cationic rhodamine 123, it is transported into mitochondria and exhibits a green fluorescence. It is therefore a marker for cellular ROS. After 15 min incubation in the dark at room temperature, ROS production was stopped by addition of 100 µL ice-cold PBS. DHR-123 oxidation was assessed by fluorescence measurement at 500/536 nm with a Tecan Infinite pro 200 microplate reader (Tecan, Männedorf, Switzerland).

### 2.7. Mitochondrial DNA Copy Number

HL60 cells (2 × 10^5^) were treated with MAA and hemin as mentioned above. After 24 h of treatment, total DNA was extracted from treated HL60 cells and purified using the Qiagen DNeasy Blood & Tissue Kit (Qiagen, Hombrechtikon, Switzerland). Purity and quantity of the obtained DNA were assessed with a NanoDrop 2000 photometer (Thermo Scientific, Wohlen, Switzerland). Real-time PCR was performed in triplicate using SYBR green (Roche Diagnostics, Rotkreuz, Basel) and specific primers (Microsynth AG, Balgach, Switzerland) for mitochondrial DNA: ND1 (forward: 5′-ATGGCCAACCTCCTACTCCT-3′, reverse: 5′-CTACAACGTTGGGGCCTTT-3′), and for nuclear: DNA, 36B4 (forward: 5′-GGAATGTGGGCTTTGTGTTC-3′, reverse: 5′-CCCAATTGTCCCCTTACCTT-3′), were used. The mitochondrial copy number was calculated according to Quiros et al. [31].

### 2.8. Mitochondrial Membrane Potential

The mitochondrial membrane potential (Δψm) was determined in 2 × 10^5^ HL60 cells using TMRM. TMRM is a cationic fluorescent probe, which accumulates within mitochondria depending on their Δψm. HL60 cells were treated with MAA and hemin as described above. After 24 h incubation, the cells were washed with PBS and resuspended in 100 µL PBS containing 100 nM TMRM in a clear v-bottom 96-well microplate (Corning, Bedford, Massachusetts, USA). After 30 min incubation in the dark, the cells were washed and resuspended with PBS. As positive control 20 μM FCCP was added to the cells 15 min before measurement. Δψm was measured by flow cytometry with a CytoFLEX flow cytometer (Beckmann Coulter, Indianapolis, IN, USA) and data were assessed using FlowJo software 10.08 (Tree Star, Ashland, OR, USA).

### 2.9. Cytotoxicity by MAA/Hemin for HL60 Cells Cultured with Galactose

In order to investigate whether MAA/hemin caused cytotoxicity by impairment of glycolysis, we compared the effect of MAA/hemin on cells with and without ATP generation by glycolysis. In the presence of glucose, mammalian cells can produce ATP by (mitochondrial) oxidative phosphorylation and by glycolysis. In the presence of galactose, cells cannot generate ATP by glycolysis but have to rely on mitochondrial ATP production. Cellular oxidation of galactose to pyruvate via glycolysis yields no net ATP, which is why the cells are forced to rely on oxidative phosphorylation to generate enough ATP for survival. [32]. We therefore pre-incubated HL60 cells in RPMI-1640 medium containing galactose (10 mM) instead of glucose for 4 h. After pre-incubation in galactose, the cells were treated in galactose medium with MAA and hemin as described above for glucose. After 24 h of treatment, cellular ATP content as marker for energy metabolism and adenylate kinase release as marker for membrane toxicity were assessed.

### 2.10. Effect of MAA/hemin on Glycolysis

The effect of MAA and hemin on glycolysis was assessed by measuring the lactate content in the supernatant and by measuring the extracellular acidification rate (ECAR) using the Seahorse XF 96 analyzer (Seahorse Bioscience, Agilent Technologies). The lactate concentration in the supernatant of cell incubations (2 × 10^5^ cells, final volume of the assay 100 µL) was determined after 8, 16, and 24 h of exposure to the toxicants by an enzymatic assay as described previously [33,34].

For the determination of the ECAR, 2 × 10^5^ HL60 cells were incubated with MAA and hemin with or without addition of the test compounds and 1 mM NAC for 24 h. After the incubation, the cells were washed in PBS and resuspended in sterile assay buffer without glucose (unbuffered RPMI-1640 without glucose supplemented with 1 mM sodium pyruvate, pH 7.4) and plated (75,000 cells/well) on a Cell-Tak (Corning, Bedford, MA, USA) coated XF 96-well cell culture microplate (Seahorse Bioscience, Agilent Technologies). The plate was incubated for 1 h in a CO_2_ free incubator at 37 °C. ECAR was determined using the extracellular flux analyzer (Seahorse Bioscience). First, the basal acidification was determined by measuring the ECAR before addition of any activators or inhibitors. Glucose-dependent ECAR was determined by injection of glucose at a final concentration of 10 mM. Then, maximal ECAR was determined after injection of oligomycin to block oxidative phosphorylation at a final concentration of 1 µM. Next, non-glycolytic ECAR was determined by injection of 2-DG at a final concentration of 62.5 mM. Glycolytic ECAR was determined by subtracting non-glycolytic ECAR from maximal glycolytic capacity.

### 2.11. GSH Content of HL60 Cells

We analyzed the cellular GSH content by liquid chromatography tandem mass spectrometry (LC-MS/MS). As a control, we treated HL60 cells with 100 µM L-buthionine sulfoximine (BSO), which inhibits the GSH biosynthesis [33]. GSH ammonium salt-d5 (~10 mM, in H_2_O) was used as internal standard. To prevent auto-oxidation of GSH and GSH ammonium salt-d5 during the sample preparation, we alkylated the thiol group with N-ethylmaleimide (NEM) forming GS-NEM and GS-NEM-d5 [34].

Briefly, 5 × 10^5^ HL60 cells were seeded in a 24-well plate and treated with MAA and hemin with or without 1 mM NAC for 24 h. After the treatment, the cells were harvested, washed, and half of the cells were lysed with RIPA buffer to determine the protein concentration. The rest of the cells were incubated in 250 µL alkylating solution (50 mM NEM in PBS) for 30 min on ice. The samples were extracted in a ratio of 1:4 (*v*/*v*) with internal standard solution, which consisted of 500 nM GS-NEM-d5 in methanol. We kept the extracts at −20 °C for 30 min to ensure protein precipitation. After centrifugation at 3500× *g* for 10 min at 4 °C, we transferred 250 µL of supernatant into a LC-MS/MS tube. The calibration line of GS-NEM (250 µM–0.25 µM) was prepared in alkylating solution and extracted as described above. Separation and quantification of GS-NEM was performed on a Shimadzu HPLC (Kyoto, Japan) coupled to an API 4000 QTrap tandem mass spectrometer (ABSciex, Concord, ON, Canada) as described previously in detail [35].

### 2.12. Quantitative Real-Time PCR

HL60 cells were incubated with MAA and hemin in 75 cm^2^ tissue culture flasks (TPP Techno Plastic Products AG, Trasadingen, Switzerland) at a density of 2 × 10^5^ cells/mL as described before. After 24 h of incubation, mRNA was extracted and purified using the Qiagen RNeasy mini extraction kit (Qiagen, Hombrechtikon, Switzerland). Purity and quantity of the obtained mRNA were assessed with a NanoDrop 2000 (Thermo Scientific, Wohlen, Switzerland) and cDNA was synthesized from 1 μg mRNA using the Qiagen omniscript system. Real-time PCR was performed in triplicate using SYBR green (Roche Diagnostics, Rotkreuz, Basel) and specific primers (Microsynth AG, Balgach, Switzerland) for hexokinase II (HKII) (forward: 5′-TGCCTGGCTAACTTCATGGA-3′, reverse: 5′-AAGTCCCCTCTCCTCTGGAT-3′), fructose-6-phosphat-kinase (PFK) (forward: 5′-GCTGATCATCGGTGGATTCG-3′, reverse: 5′-GGTCTCGATGATGAACACGC-3′), and pyruvate kinase (PK) (forward: 5′-CTACACCAACATCATGCGGG-3′, reverse: 5′-AGGTAGGGAGGGTCAGGAAT-3′). Real-time PCR was performed using the ViiA7 system (Thermo Fisher Scientific, Ecublens, Switzerland). Relative quantities of specifically amplified cDNA were calculated with the comparative threshold cycle method.

### 2.13. Western Blot Analysis

HL60 cells were incubated with MAA and hemin in 75 cm^2^ tissue culture flasks (TPP Techno Plastic Products AG, Trasadingen, Switzerland) at a density of 2 × 10^5^ cells/mL as outlined above. After 24 h of incubation, the cells were lysed in RIPA buffer (150 mM sodium chloride, 1.0% Triton X-100, 0.5% sodium deoxycholate, 0.1% sodium dodecyl sulphate, 50 mM Tris, pH adjusted to 8.0). Then, the samples were centrifuged at 500× *g* for 10 min at 4 °C and the supernatant was collected. Protein content of each sample was determined using the BCA Protein Assay Kit (Pierce, Thermo Scientific, Rockford, IL, USA). For protein separation, the supernatant was applied on 4–12% bis-tris polyacrylamide gels (Invitrogen, Basel, Switzerland) and the samples were run under reducing conditions. Afterwards, the proteins were transferred to 0.2 µm nitrocellulose membranes (Bio-Rad Laboratories AG, CA, USA). Subsequently, the membranes were blocked with 5% nonfat dry milk in PBS (Gibco, Paisley, UK) containing 0.1% Tween-20 (Sigma-Aldrich, St. Louis, MO, USA) (PBS-Tween) for 1 h at room temperature. Afterwards, the membranes were incubated overnight with a primary antibody (Abcam, Cambridge, UK) for HKII (ab209847) and GAPDH (ab8245) diluted 1:5000 and for PFK (ab154804) and PK (ab207450) diluted 1:2000 in blocking buffer. The next day, the membranes were incubated for 1 h with the corresponding secondary antibody (Santa Cruz Biotechnology, Santa Cruz, CA, USA) diluted 1:2000 in 5% nonfat milk in PBS-Tween. Afterwards, the membranes were washed and the immunoreactive bands were developed using enhanced chemiluminescence solution (GE Healthcare, Buckinghamshire, UK). Chemiluminescent images were scanned, and band intensities analyzed using a Witec Fusion pulse imaging system (Witec, Sursee, Switzerland). To correct for loading differences, the scanning units obtained for the test proteins were divided by the scanning units obtained for the housekeeping protein GAPDH.

### 2.14. Statistics

Data are presented as the mean ± SEM from at least three independent experiments. All statistical analyses were performed using GraphPad Prism 8.1.2 (GraphPad Software, La Jolla, CA, USA). Differences between many groups were tested by one-way ANOVA followed by Bonferroni’s multiple comparison tests to localize significant results in the ANOVA. Differences within two groups were tested by an unpaired *t*-test. A *p*-value < 0.05 was considered to be a significant difference.

## 3. Results

### 3.1. Time Course of MAA/hemin Toxicity in HL60 Cells

The aim of the current study was to assess how MAA and MAA/hemin affects energy metabolism in HL60 cells, in particular the generation of ATP. For that, we first studied the effect of MAA and MAA/hemin on the cellular ATP content and membrane toxicity over time. We exposed HL60 cells to MAA and MAA/hemin and assessed the cellular ATP content and membrane toxicity every 4 h over 24 h.

As shown in Figure 1A, the cellular ATP content started to decrease after 8 h of treatment, while membrane toxicity was observed only after 12 h (Figure 1B). In comparison, hemin alone started to decrease the cellular ATP at 12 h and was less toxic than MAA/hemin between 8 and 24 h. MAA alone did not affect the cellular ATP content. Membrane toxicity was observed from 12 h on for both MAA/hemin and hemin alone, but the combination MAA/hemin was always more toxic than hemin (Figure 1B). MAA alone exhibited no membrane toxicity.

### 3.2. Mitochondrial Involvement in MAA/hemin Toxicity

To investigate a possible role of mitochondria for the observed drop in cellular ATP, we studied the effect of MAA and MAA/hemin on cellular oxygen consumption and on the accumulation of mitochondrial superoxide (O_2_^•−^) and cellular ROS.

As shown in Figure 2A and as quantified in Appendix A, in comparison to control incubations, the addition of MAA, hemin or the combination MAA/hemin did not affect ATP-coupled respiration, but hemin alone and MAA/hemin increased maximal respiration. In order to study the possible role of reactive metabolites, we repeated these experiments in the presence of *N*-acetylcysteine (NAC) (Figure 2B and Appendix A). Hemin alone and the combination MAA/hemin increased ATP-coupled respiration compared to control values (but not compared to the same incubations without NAC) and NAC significantly reduced the increase in maximal respiration observed in the presence of hemin or MAA/hemin in the absence of NAC.

We also assessed whether MAA and/or MAA/hemin affected cellular or mitochondrial ROS production. It is well-established that inhibition of the mitochondrial electron transport chain is associated with an increase in mitochondrial superoxide and cellular ROS accumulation [36]. Hemin slightly increased the mitochondrial superoxide radical (O_2_^•−^) content and this effect was not accentuated by the addition of MAA (Figure 2C). MAA alone, potentially due to its antioxidant capacity [37], decreased the mitochondrial superoxide content in otherwise untreated cells by trend. We also assessed the cellular ROS generation by HL60 cells exposed to MAA with or without hemin for 24 h using DHR-123 as a probe. As shown in Figure 2D, MAA significantly decreased the cellular ROS content independently of hemin, whereas hemin increased the cellular ROS content by trend.

To further investigate a possible effect of MAA with or without hemin on mitochondria, we assessed the mitochondrial DNA copy number of HL60 cells treated for 24 h. As shown in Appendix A, neither hemin nor MAA alone or the combination of MAA/hemin significantly affected the mitochondrial DNA copy number.

We also assessed whether MAA/hemin affected the mitochondrial membrane potential (Δψm) in HL60 cells. A decrease in Δψm would reflect impaired mitochondrial function and integrity, in particular also the function of the mitochondrial electron transport chain. As shown in suppl. Appendix A, MAA alone did not affect Δψm and hemin alone decreased Δψm significantly by 7% (6.25 µM) or 11% (12.5 µM). The effect of hemin was not significantly increased by MAA.

This minimal effect on the Δψm suggested that MAA/hemin would have only a small effect on cellular oxygen consumption, which reflects mainly the activity of the mitochondrial electron transport chain. As shown in Figure 2A, in comparison to control incubations, the addition of MAA, hemin or the combination MAA/hemin even increased the cellular oxygen consumption rate by HL60 cells, excluding inhibition of the mitochondrial electron transport chain as a reason for the drop in the cellular ATP content.

These results showed that MAA does not impair ATP coupled or maximal respiration independently of the presence of hemin and that MAA can act as an antioxidant. The effect of hemin on maximal respiration appeared to be associated with the production of reactive metabolites that stimulated the mitochondrial electron transport chain. This effect was not further investigated since this was not the primary aim of this project.

### 3.3. Impact of Culture Conditions on MAA/Hemin Toxicity

A possibility to differentiate between ATP production by mitochondria and by glycolysis is by culturing cells in the presence of glucose or in the presence of galactose. Since galactose cannot be used for glycolysis, cells are forced to generate ATP by the mitochondrial respiratory chain [32]. In the presence of glucose, ATP generation by both glycolysis and the mitochondrial respiratory chain is possible.

In HL60 cells incubated with glucose (Figure 3A and Appendix A) MAA/hemin decreased the cellular ATP content more than hemin alone. In comparison, in HL60 cells cultured in with galactose, MAA/hemin also decreased the cellular ATP pool, but to the same extent as hemin (Figure 3B and Appendix A). Accordingly, membrane toxicity, as illustrated by adenylate kinase release, was more accentuated after treatment with MAA/hemin in glucose-containing medium compared to hemin alone (Figure 3C). In the galactose medium, the observed cytotoxicity of MAA/hemin was still increased compared to hemin alone, but this increase was markedly less than in the medium with glucose (Figure 3D). Hemin alone was more toxic in the presence of galactose than of glucose. MAA alone showed no significant effect on the cellular ATP pool and membrane intactness under both culture conditions.

These results showed that hemin is a mitochondrial toxicant, as indicated by the more accentuated drop in the cellular ATP content in cells cultured with galactose compared to glucose. Taking into account the results of Figure 2A, this effect had to be directed on mitochondrial ATP generation and/or ATP consumption. In contrast, MAA selectively inhibited glycolysis in the presence of hemin, since it only decreased the cellular ATP content in cells cultured with glucose. An effect of hemin on glycolysis was not shown but cannot be excluded by these investigations.

### 3.4. Effect of MAA/Hemin on Glycolysis

In order to confirm the effect of MAA on glycolysis and to investigate whether hemin also decreases glycolysis, we studied the effect of MAA/hemin on glycolysis directly. First, we determined the effect of MAA/hemin on the lactate production by HL60 cells. As shown in Figure 4A, the lactate concentration in the cell supernatant increased linearly with time under all incubation conditions. The respective velocities (mean ± SEM) were 92.5 ± 8.5 pmol/10^6^ cells/min for control incubations, 51.3 ± 8.1 pmol/10^6^ cells/min in the presence of hemin (*p* < 0.05 vs. control) and 33.7 ± 4.4 pmol/10^6^ cells/min in the presence of MAA/hemin (*p* < 0.05 vs. control and *p* = 0.105 vs. hemin). These results indicated that the addition of MAA accentuated the decrease in lactate production from glucose by hemin in HL60 cells.

In order to confirm these findings, we analyzed the glycolytic capacity of HL60 cells by determining the extracellular acidification rate (ECAR) in the presence of glucose using the Seahorse XF 96 analyzer. As shown in Figure 4B and quantified in Table 1, the extracellular acidification rate by HL60 cells in the presence of glucose and the maximal glycolytic capacity after addition of oligomycin were not affected by MAA alone. However, the addition of MAA to hemin enhanced the impairment of the acidification rate observed with hemin alone.

In order to investigate the possibility that electrophilic MAA radicals are involved in the toxicity of MAA/hemin, we repeated the experiment in the presence of the radical scavenger NAC. As shown in Figure 4C and quantified in Table 1, the effect of hemin on glycolysis was not affected by NAC, whereas the additional effect of MAA in MAA/hemin was blunted.

Since the addition of NAC blunted the inhibitory effect of MAA/hemin on glycolysis, we were interested whether this was a direct effect of NAC or an indirect effect of NAC via an increased cellular GSH content. As shown in Figure 4D, NAC and MAA did not increase the cellular GSH content, whereas hemin increased the GSH content significantly. These results suggested a direct effect of NAC on the toxicity of MAA/hemin by scavenging formed reactive MAA intermediates.

The findings in Figure 4 confirmed that MAA/hemin impaired glycolysis and suggested that reactive metabolites of MAA formed in the presence of hemin are involved in the toxic effects associated with MAA/hemin.

### 3.5. Molecular Mechanism of the Inhibition of Glycolysis by MAA/Hemin

ATP generation by glycolysis involves ten enzymes, which transform glucose to pyruvate and lactate [38]. Within this process, hexokinase, phosphofructokinase, and pyruvate kinase are regulatory enzymes. In order to find a possible mechanism for our observation that MAA/hemin and hemin impair glycolysis, we determined the mRNA and protein expression of these enzymes in HL60 cells treated with MAA, hemin or MAA/hemin. The mRNA expression of hexokinase and phosphofructokinase was not significantly altered by MAA, hemin, or MAA/hemin (Figure 5A–C). However, hemin and MAA/hemin numerically reduced the mRNA expression of pyruvate kinase, a cytosolic enzyme with a prominent position in the formation of pyruvate. In agreement with this finding, the protein expression of pyruvate kinase was significantly reduced by MAA/hemin and, to a smaller extent, also by hemin (Figure 5F). In comparison and in agreement with mRNA expression, the protein expression of hexokinase II and phosphofructokinase was not decreased by MAA/hemin or hemin (Figure 5D,E). Interestingly, MAA alone and, less accentuated, also the combination MAA/hemin increased the protein expression of hexokinase II (Figure 5D). Hence, our data suggest that MAA/hemin impairs glycolysis by HL60 cells by reducing the protein content of pyruvate kinase, which catalyzes an essential step of glycolysis.

## 4. Discussion

The current study shows that MAA enhanced the toxic effect of hemin on glycolysis but not on mitochondrial function and that MAA alone did not impair the ATP metabolism of HL60 cells. The radical scavenger NAC reduced the effect of MAA added to hemin on glycolysis but had no such effect on the toxicity of hemin alone, confirming that electrophilic MAA intermediates play a role in MAA-associated toxicity.

The observation that reactive MAA intermediates formed by MAA/hemin interfere with glycolysis, causing a drop in ATP and subsequent cell death, is a possible explanation why in patients with metamizole myelotoxicity neutrophils are more frequently affected than other cell types. Kramer et al. showed that, among all blood cells, neutrophils have the lowest oxygen consumption rate (OCR)/ECAR ratio, indicating that neutrophils rely to a higher extent on glycolysis for ATP production than the other blood cells [39]. Regarding the main function of neutrophils, host defense during bacterial infections under hypoxic conditions, ATP production by glycolysis is preferable to oxidative phosphorylation [40,41]. In contrast, monocytes and lymphocytes produce ATP predominantly by oxidative phosphorylation [39,42], rendering these cells potentially less susceptible to the toxicity of MAA/hemin. However, upon activation in case of inflammation, macrophages and lymphocytes can metabolically switch to glycolysis in order to adapt to hypoxia [40]. Indeed, lymphopenia can be observed in patients with metamizole-induced neutropenia [21,43,44], supporting the notion that MAA/hemin is more toxic for cells generating ATP by glycolysis than by oxidative phosphorylation.

Since erythrocytes have no mitochondria, they have to rely solely on glycolysis for ATP generation [45] and could therefore be a target for MAA/hemin toxicity. In support of this assumption, anemia possibly associated with the use of metamizole has been reported in patients with metamizole-induced neutropenia [12,22]. However, anemia is not as frequent as would be expected in patients with metamizole-induced neutropenia, when ATP generation by glycolysis is considered to be an important risk factor for metamizole-induced cytotoxicity. Since reticulocytes, the direct erythrocyte precursors, still contain mitochondria [45], only mature erythrocytes are a potential target of this type of toxicity. However, the number of mature erythrocytes is small in bone marrow. The fact that metamizole exerts its cytotoxicity in the bone marrow and not in the peripheral blood [23,26] may explain why metamizole has only a minor effect on erythropoiesis. On the other hand, impairment of glycolysis in mature erythrocytes can cause hemolysis [46,47]. By this mechanism, the free heme may increase in the bone marrow, which could cause the generation of reactive MAA intermediates and subsequent toxicity on neutrophil granulocyte precursors [23].

Our results suggest that the observed impairment of glycolysis in HL60 cells by MAA/hemin can be explained by reduced expression of pyruvate kinase (PK) protein. Pyruvate kinase is expressed in mammals as four different isozymes PKL, PKR, PKM1, and PKM2, whereby PKM2 is expressed in leukocytes [45]. It catalyzes the final, irreversible, and rate-limiting step of glycolysis by transferring the phosphate group of phosphoenolpyruvate to ADP, yielding pyruvate and one of the two ATP molecules generated by glycolysis [48]. Taking into account these properties, pyruvate kinase represents a key regulator enzyme of glycolysis [45]. Long-term regulation of pyruvate kinase activity is achieved by modulation of the transcription by hormones such as insulin, nutrients, and mitogenic signaling pathways, which control protein expression [49,50]. As shown in Figure 5, hemin and MAA/hemin decreased the mRNA expression of pyruvate kinase numerically without reaching statistical significance, whereas the effect on protein expression was strong for MAA/hemin and hemin, but more accentuated for MAA/hemin. It is therefore possible that MAA/hemin and, to a lesser extent, hemin alone affected pyruvate kinase protein expression, either by impairing the translation or by reducing the protein stability. Reduced protein stability could be explained by the formation of electrophilic, reactive metabolites from MAA [23], and by the oxidative capacity of hemin [51], which could damage amino acids of pyruvate kinase, eventually leading to the degradation of the enzyme [52]. However, the protein expression of other key enzymes of glycolysis, hexokinase II and phosphofructokinase, was not affected by MAA/hemin. This difference compared to pyruvate kinase cannot be explained by the current study but may be due to the different primary and secondary structures of these enzymes. It is also possible that MAA/hemin only interferes with the PKM2 isozyme of pyruvate kinase, leaving other isozymes, such as the PKR isozyme in erythrocytes, unimpaired.

The increase in basal and maximal cellular oxygen consumption by hemin and MAA/hemin excluded the possibility that the decrease in the cellular ATP by hemin was due to the inhibition of the electron transport chain. On the other hand, the decrease of the cellular ATP pool in the presence of galactose clearly indicated that hemin decreased the mitochondrial production of ATP and/or increased mitochondrial ATP consumption. Uncoupling of oxidative phosphorylation, which would increase mitochondrial ATP consumption, can be excluded, since oxygen consumption after the addition of oligomycin and before addition of the uncoupler FCCP (proton leak in Figure 2B), which would be increased in the presence of an uncoupler, was not different between the incubations containing hemin and control incubations (Figure 2B). This suggests that hemin inhibits mitochondrial ATP synthesis, possibly by impairing the function of the mitochondrial ATP synthase. The observed increase in maximal oxygen consumption by HL60 cells in the presence of hemin and MAA/hemin could represent a compensatory reaction in view of the inhibition of the mitochondrial ATP synthesis.

In conclusion, the current investigations support the hypothesis derived from our previous studies [23,25] that the depletion of the cellular ATP pool is important for the toxicity of MAA/hemin on HL60 cells. MAA itself is not toxic. In the presence of hemin, the toxic principle of MAA is based on the formation of reactive metabolites, which inhibit glycolysis but not the mitochondrial electron transport chain.

## Figures and Tables

**Figure 1 biomedicines-08-00212-f001:**
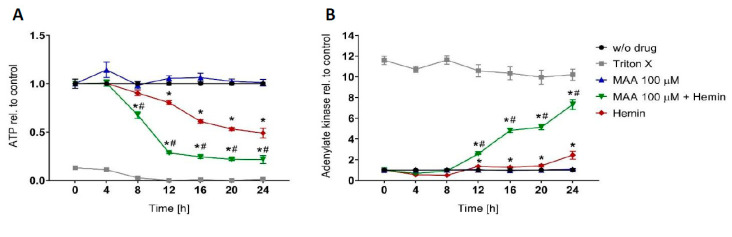
Time dependency of the effect N-methyl-aminoantipyrine (MAA) and/or hemin on the ATP content and plasma membrane intactness of HL60 cells. HL60 cells were treated with the toxicants for 24 h. (**A**) Cellular ATP content determined using the CellTiter-Glo^®^ luminescent assay. (**B**) Intactness of the plasma membrane of HL60 cells determined by the release of adenylate kinase into the supernatant. *n* = 3 independent experiments in triplicate presented as the mean ± SEM. * *p* < 0.05 vs. value at time 0 h. # *p* < 0.05 vs. the corresponding value of incubations with hemin only.

**Figure 2 biomedicines-08-00212-f002:**
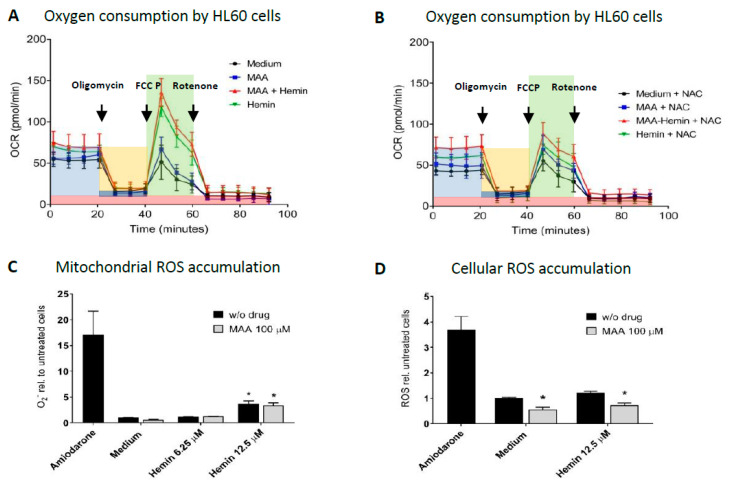
Effect of MAA and/or hemin on cellular oxygen consumption and on mitochondrial and cellular ROS generation. HL60 cells were cultured in the presence of 10 mM glucose and treated with the toxicants for 24 h. (**A**) Effect on cellular oxygen consumption. Oligomycin (1 µM) was added to block mitochondrial ATP production and carbonyl cyanide-4-(trifluoromethoxy)phenylhydrazone (FCCP) (2 µM) to uncouple oxidative phosphorylation. Highlighted in light blue: basal respiration; highlighted in orange: ATP coupled respiration; highlighted in dark blue: proton leak; highlighted in green: maximal respiratory capacity; highlighted in red: non-mitochondrial oxygen consumption; MAA 100 µM, hemin 12.5 µM. (**B**) Effect on cellular oxygen consumption in the presence of the radical scavenger N-acetylcysteine (NAC; 1 mM). (**C**) Effect on mitochondrial superoxide anion generation using MitoSOX. Amiodarone (50 µM) was used as a positive control. (**D**) Effect on cellular ROS production assessed using DHR-123. Amiodarone (50 µM) was used as a positive control. Results are presented related to vehicle-treated control incubations. *n* = 3 independent experiments in triplicate presented as the mean ± SEM. * *p* < 0.05 vs. vehicle-treated control incubations.

**Figure 3 biomedicines-08-00212-f003:**
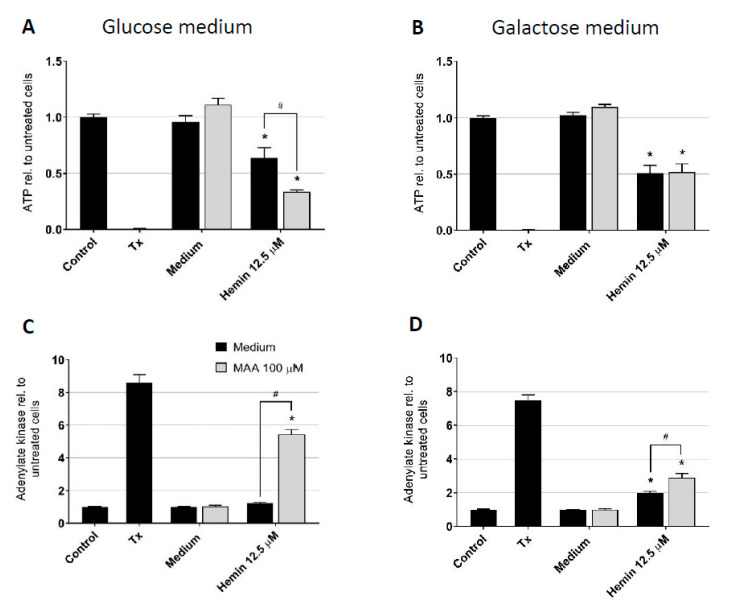
Influence of the culture conditions on the effects of MAA and/or hemin on the cellular ATP content and plasma membrane integrity of HL60 cells. HL60 cells were treated with the toxicants for 24 h. Cells were cultured in the presence of 10 mM glucose or in the presence of 10 mM galactose in order to force the cells to mitochondrial ATP generation. (**A**) Cellular ATP content determined using the CellTiter-Glo^®^ luminescent assay in cells cultured in glucose. (**B**) Intactness of the plasma membrane of HL60 cells determined by the release of adenylate kinase into the supernatant in cells cultured in glucose. (**C**) Cellular ATP content determined using the CellTiter-Glo^®^ luminescent assay in cells cultured in galactose. (**D**) Intactness of the plasma membrane of HL60 cells determined by the release of adenylate kinase into the supernatant in cells cultured in galactose. Results are presented related to vehicle-treated control incubations. Tx = 0.1% Triton X used as positive control. *n* = 3 independent experiments in triplicate presented as the mean ± SEM. * *p* < 0.05 vs. vehicle-treated control incubations. # *p* < 0.05 vs. the corresponding value of incubations 0.05 with hemin only.

**Figure 4 biomedicines-08-00212-f004:**
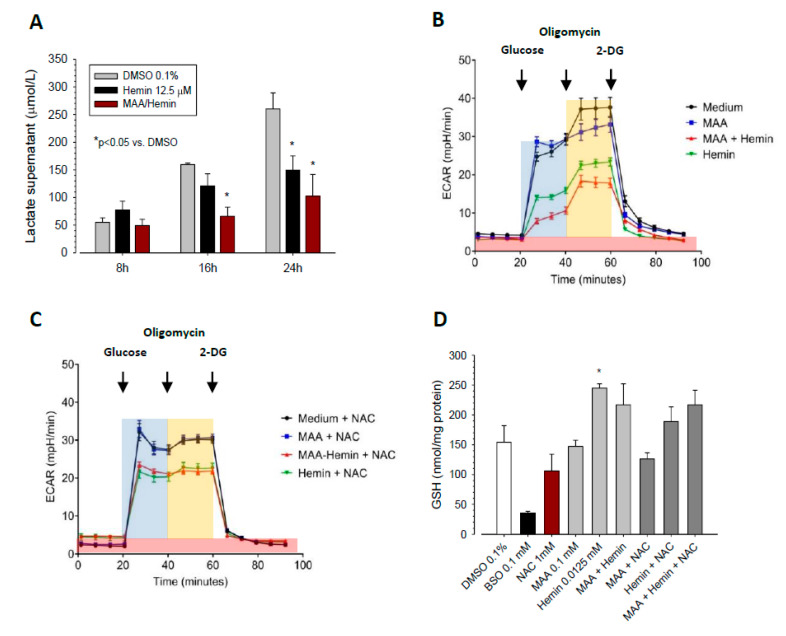
Effect of MAA and/or hemin on lactate production, extracellular acidification (ECAR) and the glutathione (GSH) content of HL60 cells. HL60 cells were cultured in 10 mM glucose and treated with the toxicants for 24 h. (**A**) Cellular production of lactate. The quantification is given in Table 1. (**B**) ECAR in the supernatant of HL60 cell treated with the toxicants as a marker of glycolysis. Glucose (10 mM), oligomycin (1 µM), and 2-deoxyglucose (2-DG) (62.5 mM) were added as indicated in the figure. Highlighted in light blue: glycolysis; highlighted in orange: maximal glycolytic capacity, highlighted in red: non-glycolytic acidification. (**C**) Acidification rate in the presence of the different toxicants and 1 mM N-acetylcysteine (NAC). (**D**) GSH content in HL60 cells after exposure to the toxicants for 24 h. *n* = 3 observations in triplicate presented as the mean ± SEM. * *p* < 0.05 vs. vehicle-treated control incubations (medium containing 0.1% DMSO); MAA 100 µM, hemin 12.5 µM.

**Figure 5 biomedicines-08-00212-f005:**
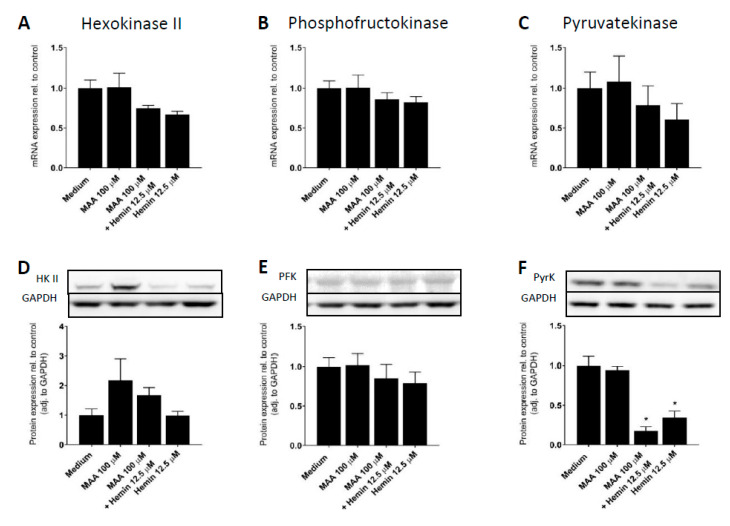
Effect of MAA and/or hemin on mRNA and protein expression of key enzymes of glycolysis in HL60 cells. HL60 cells were treated with the toxicants for 24 h. mRNA expression was determined by quantitative real-time PCR and protein expression by Western blotting. (**A**–**C**) mRNA expression of hexokinase II, phosphofructokinase and pyruvate kinase. (**D**–**F**) protein expression of hexokinase II (HKII), phosphofructokinase (PFK), and pyruvate kinase (PyrK). *n* = 3 blots per column. Values are given relative to control incubations treated with vehicle only and are presented as mean ± SEM. * *p* < 0.05 vs. vehicle-treated control incubations (medium).

**Table 1 biomedicines-08-00212-t001:** Quantification of glycolysis by HL60 cells. The corresponding traces are shown in Figure 4B (no NAC) and Figure 4C (plus NAC 1 mM). HL60 cells were treated with the toxicants for 24 h. Glycolysis was determined as the extracellular acidification rate (ECAR) in the supernatant as the drop in pH (expressed as mpH) per min in the presence of 10 mM glucose (glucose-dependent glycolysis) or glucose and 1 µM oligomycin (maximal glycolysis). Values are presented as the mean ± SEM of *n* = 4 independent observations in triplicate. * *p* < 0.05 vs. vehicle-treated control incubations (DMSO 0.1%). ^+^
*p* < 0.05 vs. hemin 12.5 µM.

	No *N*-Acetylcysteine (NAC)	
	Glucose-Dependent Glycolysis	Maximal Glycolysis
Control (DMSO 0.1%)	21.0 ± 2.3	31.5 ± 4.9
MAA 100 µM	22.5 ± 2.6	25.9 ± 2.6
Hemin 12.5 µM	10.9 ± 0.4 *	19.1 ± 0.8 *
MAA + hemin	4.9 ± 1.5 *^,+^	13.5 ± 1.1 *^,+^
	**Plus *N*-acetylcysteine (NAC) 1 mM**	
	**Glucose-Dependent Glycolysis**	**Maximal Glycolysis**
Control (DMSO 0.1%)	25.9 ± 1.9	26.6 ± 1.1
MAA 100 µM	25.9 ± 3.4	27.0 ± 1.5
Hemin 12.5 µM	17.3 ± 2.6	19.1 ± 1.1 *
MAA + hemin	18.4 ± 1.5	17.6 ± 0.4 *

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
