# Peer review of "Reactive Metamizole Metabolites Enhance the Toxicity of Hemin on the ATP Pool in HL60 Cells by Inhibition of Glycolysis"

_biomedicines, 2020, doi:10.3390/biomedicines8070212_

Round 1

Reviewer 1 Report

A slight comments are marked directly in the attached manuscript.

Author Response

A slight comments are marked directly in the attached manuscript.

Line 316 and line 318: “DNA” was missing in “mitochondrial DNA copy number”.

Answer: We appreciate the advice and added the missing word.

Reviewer #1: Line 415: really, is pyuvate kinase mitochondrial enzyme?? Pyruvate is produced in cytosol!

Answer: We thank the reviewer for the comment and agree that pyruvate kinase is no mitochondrial enzyme. We adapted the sentence accordingly to “…pyruvate kinase, a cytosolic enzyme with a prominent position in the formation of pyruvate”.

Reviewer 2 Report

The manuscript is very interesting, well written and the initial hypothesis  that the depletion of the cellular ATP pool is important for the toxicity of MAA/hemin on HL60 cells was clearly demonstrated.

I just suggest to  authors to take in consideration  to perform or to comment the opportunity to repeat the experiment described in Figure 3 comparing HL60 cultured in normal condition vs hypoxya.

Further, could be interesting to perform a quick experiment using JC1 to detect an eventual toxic effect that induce change of mitochondrial membrane potential after the MAA/hemin treatment. 

Author Response

The manuscript is very interesting, well written and the initial hypothesis that the depletion of the cellular ATP pool is important for the toxicity of MAA/hemin on HL60 cells was clearly demonstrated.

I just suggest to authors to take in consideration to perform or to comment the opportunity to repeat the experiment described in Figure 3 comparing HL60 cultured in normal condition vs hypoxya.

Answer: We agree with the Reviewer that it would be interesting to see whether MAA/hemin toxicity increases with increased glycolysis due to reduced oxygen levels. However, setting up this experiment would be beyond the scope of the available time for the current revision. Much more time would be needed to adjust for the optimal oxygen deprivation without inducing already further cellular rescue pathways against hypoxia, which might influence with the obtained results. We therefore propose to postpone this experiment for future investigations.

Further, could be interesting to perform a quick experiment using JC1 to detect an eventual toxic effect that induce change of mitochondrial membrane potential after the MAA/hemin treatment.

Answer: We performed mitochondrial membrane potential experiments with tetramethylrhodamine methyl ester (TMRM), which is a cell-permeant dye that accumulates in active mitochondria. We observed a minor but concentration-dependent decrease in membrane potential due to hemin, which was not accentuated by the addition of MAA. In contrast, at the lower hemin concentration tested (6.25 µM), MAA attenuated the effect of hemin on the mitochondrial membrane potential. We, therefore, concluded that reactive MAA intermediates formed by MAA/hemin do not interfere with the mitochondrial membrane potential.

We added the respective figure to the supplementary material (suppl. Fig. S3) and complemented the respective text sections of the methods section (2.8) and the results section (3.2).